# Effect of pH on Total Volume Membrane Charge Density in the Nanofiltration of Aqueous Solutions of Nitrate Salts of Heavy Metals

**DOI:** 10.3390/membranes10090235

**Published:** 2020-09-14

**Authors:** Agata Marecka-Migacz, Piotr Tomasz Mitkowski, Arkadiusz Nędzarek, Jacek Różański, Waldemar Szaferski

**Affiliations:** 1Division of Chemical Engineering and Equipment, Institute of Chemical Technology and Engineering, Poznan University of Technology, 60-965 Poznań, Poland; agata.j.marecka@doctorate.put.poznan.pl (A.M.-M.); jacek.rozanski@put.poznan.pl (J.R.); waldemar.szaferski@put.poznan.pl (W.S.); 2Department of Aquatic Bioengineering and Aquaculture, West Pomeranian University of Technology in Szczecin, 71-550 Szczecin, Poland; Arkadiusz.Nedzarek@zut.edu.pl

**Keywords:** nanofiltration, DSPM model, heavy metals, total volume membrane charge density, ceramic membrane

## Abstract

The separation efficiencies of aqueous solutions containing nitric salts of Zn, Cu, Fe or Pb at various pH in process of nanofiltration have been investigated experimentally. These results were used to obtain the total volume membrane charge densities, through mathematical modelling based on the Donnan–Steric partitioning Model. The experimentally obtained retention values of individual heavy metal ions varied between 36% (Zn^2+^ at pH = 2), 57% (Pb^2+^ at pH = 2), 80% (Fe^3+^ at pH = 9), and up to 97% (Cu^2+^ at pH = 9). The mathematical modelling allowed for fitting the total volume membrane charge density (*X_d_*), which yielded values ranging from −451.90 to +900.16 mol/m^3^ for different non-symmetric ions. This study presents the application of nanofiltration (NF) modelling, including a consideration of each ion present in the NF system—even those originating from solutions used to adjust the pH values of the feed.

## 1. Introduction

Nowadays, dynamically thriving chemical plants produce high volumes of wastewaters, and, therefore, in many cases, they are a source of polluted water containing heavy metals. In many references, Cu, Pb and Zn have been mentioned as the most dangerous heavy metals, which are produced by chemical-intensive industries at a large scale [1,2,3]. Based on the information placed on the website of the United States Environmental Protection Agency and according to The National Primary Drinking Water Regulations (NPDWR) [4], the concentration of heavy metals such as Cu, Fe, Zn or Pb in drinking water cannot exceed 1.3 mg/L, 0.3 mg/L, 5 mg/L and 0.015 mg/L, respectively. The origins of pollution with heavy metals are various; however, the sources can be related to the corrosion of household plumbing systems, the erosion of natural deposits across liquid industrial wastes in ore enrichment plants, inorganic paint factories, production involving galvanization, etc.

Current knowledge indicates that it is better to prevent than combat the effects; therefore, recent research has been directed towards methods of preventing the migration of heavy metals from industrial wastewaters to the environment at the source, rather than through their treatment later on. The removal of heavy metals from inorganic effluents can be achieved by conventional treatment processes, such as chemical precipitation, flotation, ion exchange and electrochemical deposition. These processes have significant disadvantages, which include: incomplete removal, high-energy requirements, and the production of toxic sludge [2]. Newer processes, such as adsorption on novel adsorbents (natural materials) [5], photocatalytic processes [6], electrodialysis [7] or membrane processes [8,9,10,11,12,13], appear to be more effective than traditional treatment methods [14]. When high contaminant removal is a goal, nanofiltration is generally found to be cost-effective [14,15]. However, wide industrial applications are limited by the relatively high operational costs [14]. In the last 20 years, membrane processes have gained significant attention in the field of separation processes [16,17,18]. The continuous development of new polymeric and inorganic membranes with high efficiency and selectivity as well as the improved knowledge regarding separation mechanisms allowed for the replacement of conventional techniques using membrane processes [19]. Nanofiltration (NF) is a process with low power demand in comparison to reverse osmosis or distillation, which works in the pressure range of 0.4–3 MPa, and also it does not introduce any additional ingredients that may pose problems with their removal, or affect the purity of the product [17,19,20,21].

One of the most important features of nanofiltration membranes is their ability to separate ions from water. The NF process can recover metallic ions, or at least retain them, and it can be used to concentrate solutions containing multivalent salts or to fractionate salts based on the different charge densities and hydrated sizes of the ions [22,23]. Examples of applications of ions and hardness recovery with use of NF processes are presented in many reports [16,24,25,26]. With the growing interest in NF as a separation technique for a wide range of applications, even under harsh conditions, ZrO_2_ or TiO_2_ membranes are increasingly preferred due to their high chemical, thermal and mechanical resistance [27,28] and easy interaction between metallic species and ceramic materials [25]. Therefore, the use of ceramic membranes in aggressive systems, including extreme pH values, is recommended [29,30,31].

In general, transport during NF depends on diffusion, convection and electrostatic interactions [32]. For a charged compound, both steric hindrance and electrostatic interactions are responsible for rejection [33,34]. Another important parameter in the transport and interpretation of retention is the membrane charge present along the surface of a membrane and also through the pores [35]. A strong charge present at the membrane surface has a crucial effect on the ion retention of the membrane [36]; unfortunately, the experimental determination of the membrane charge, which could explain ion transport through a NF membrane, is challenging. Therefore, a modelling-based approach has been published [23]. However, there is no experimental technique which would enable the quantification of the membrane charge value in direct way, especially during separation. Nowadays, the only possible way is to use streaming potential techniques [37,38,39]. As a result, zeta-potential values are obtained, while such measurement methods require a sample in a flat, powder or even fibre form, which requires the destruction of a membrane. Therefore, the authors of this study postulate using mathematical modelling to determine the total volume membrane charge density and correlate the pH of separated solutions, which would help in the assessment of membrane performance.

In the mathematical modelling of NF, three groups of models describing transport across a membrane can be distinguished. The first group of models is derived from irreversible thermodynamics and considers the membrane as a black box. The other two groups of models additionally take into account the properties of the membrane and are divided into: solution-diffusion and pore-flow models [19]. Over the last few decades many transport models have been proposed such as the Steric-Hindrance-Pore model, Electric-Steric-Hindrance-Pore model, Teorell–Mayer–Sievers model, Frictional model, or Space-Charge model [19,40,41]. More recently, Nair et al. [42] determined membrane transport parameters and effective pore size with the Spiegler-Kedem model and the Steric-Hindrance-Pore model. In 2019 Nair et al. [43] explained the effect of pH on flux variation with the use of the Spiegler–Kedem and Steric-Hindrance-Pore models. Bowen et al. [44,45,46,47] proposed the Donnan–Steric partitioning model (DSPM), which has also been used by others [48,49,50,51,52,53,54] with fairly good results. Xu et al. [55] investigated temperature influences on the retention of fourteen kinds of pharmaceuticals and personal care products by NF membranes and predicted their performances at given feed temperatures with the use of refined DSPM and a Dielectric-Exclusion model incorporated with temperature functions. Kingsbury and co-authors [56] used the solution-diffusion model as a common framework to compare the permeability, partition and diffusion coefficients, water permeance, and salt rejection of twenty commercial ion exchange membranes. Despite the modelling approaches presented above, the novel computational methodology was developed by Rall et al. [57], who integrated accurate physical models of ion transport—valid on the nano-scale—into the large-scale superstructure optimization of the membrane. Nevertheless, none of these models are fully predictive, due to the difficulties associated with the identification of certain model parameters [58].

In order to predict the separation performance, it is important to evaluate the membrane charge density in well-defined solutions. To the best of our knowledge, there is no paper presenting the volume charge membrane densities obtained with the DSPM model correlated with values of pH. Therefore, the authors aim to consider each ion and water in modelling, to therefore provide values of the total volume membrane charge densities—along with their correlation to the pH of separated solutions—through mathematical modelling. As a case study, the nanofiltration of aqueous solutions of nitric salts of Cu^2+^, Zn^2+^, Fe^3+^ and Pb^2+^, at various pH values, was studied experimentally and with the described DSPM model. 

## 2. Materials and Methods

All experiments were conducted using the experimental set-up schematically presented in Figure 1, equipped with 19-channel ceramic Al_2_O_3_/TiO_2_ nanofiltration membranes (Inopor, Germany) with the following characteristics: cut-off at 450 Da, porosity of 0.3, membrane active layer thickness 0.5 µm [59], pore radius 0.9 nm, length of 1178 mm, external diameter of 25 mm, channel diameter of 3.5 mm, and a filtration area of 0.25 m^2^. The Point of Zero Charge (PZC) of the membrane used in the NF process was equal to 6.0 ± 0.9 (reported by the manufacturer). Transmembrane pressure (TMP) was set to 0.4 MPa and the cross-flow velocity was equal to 4 m/s. The process temperature was fixed and amounted to 293 ± 1.0 K. The system operated in continuous mode. Thus, both permeate and retentate were driven to the feed reservoir to keep the concentration of the experiments constant and simulate a continuous filtration process. The steady state was usually obtained after 90 min of operation in continuous mode. More details regarding the experimental set-up can be found in [60]. After each filtration, the membrane was chemically cleaned according to the manufacturer’s recommendations, described by Nędzarek et al. [30] and Bonisławska et al. [61]: washing with 2% NaOH solution (T = 360 K, t = 40 min), rinsing with ultra-pure water, washing with 0.5% HNO_3_ solution (T = 320 K, t = 30 min), and, finally, rinsing three times with ultra-pure water. Chemical washing resulted in a performance characteristic for a clean membrane. 

Working solutions subjected to nanofiltration contained single aqueous solutions of Zn(NO_3_)_2_, Cu(NO_3_)_2_, Pb(NO_3_)_2_ and Fe(NO_3_)_3_. The initial concentration of each heavy metal ion in the solution before filtration was equal to 500 μg/dm^3^, and no precipitation was observed in all investigated systems. The retention levels of the metals were measured for the following pH values: 2.0, 4.6, 6.0, 6.9, and 9.0. The pH values were controlled using 0.1 M NaOH and 0.1 M HCl. In the feeds and permeates obtained at the respective pH values, heavy metals were traced by a cathodic stripping voltammetry method (CSV). The heavy metal retention (*R)* was calculated according to Equation (1):(1)$$R=\left(1-\frac{C_{p,i}}{C_{f,i}}\right)\cdot 100\%,$$
where *C_f,i_* is the concentration of individual ion in the feed and *C_p,i_* in the permeate solution.

### Determination of Effective Membrane Charge Density in Nanofiltration

In order to describe the ion transport through the NF ceramic membrane, the DSPM model was utilized. In comparison with other reports [44,45,46,47,48,49,50,51,52,62], the modelling in this contribution considered each ion present in the system—even ions originating from solutions used to set the desired values of pH. Such a detailed approach is innovative in terms of modelling NF processes. To date, researchers who exploited the DSPM model did not take ions originating from solutions used for regulating pH, such as NaOH or HCl, into account, or at least did not show it explicitly. The solutes in aqueous solutions dissociate, then deliver specific ionic forms to the separated system. The authors are convinced that the presence of additional ions (such as Na^+^, OH^−^ or H^+^, Cl^−^) may influence the total volume membrane charge density. In the model, it was also assumed that the concentrations of the components in the feed are constant (i.e., steady state model), that the transmembrane pressure for the entire duration of the process is constant, that the straight cylindrical pores of length are equal to the effective membrane layer thickness, and that the concentration polarization effect and fouling phenomena are negligible. The Reynolds number at the feed side was equal to 13,309 (*Re* = *w**·**d**·**ρ**/η*). The solvent viscosity values were assumed to be equal to water at a process temperature equal to 20 °C, which is (*η**_s_*) 0.00105 Pa·s. Therefore, the schematic representation of the concentration profiles is shown in Figure 2, which can be described in detail using the following set of model Equations (2)–(17).
(2)$$V_{s}=\frac{r_{p}^{2}\left(\Delta P-\Delta \pi \right)}{8\eta_{s}\Delta x}$$
(3)$$\Delta \pi =\pi_{feed}-\pi_{permeate}$$
(4)$$\pi_{feed}=\frac{RT}{\tilde{V_{w}}}\sum_{i=1}^{NoComp}x_{f,i}$$
(5)$$\pi_{permeate}=\frac{RT}{\tilde{V_{w}}}\sum_{i=1}^{NoComp}x_{p,i}$$
(6)$$\lambda_{i}=\frac{r_{s,i}}{r_{p}}$$
(7)$$\phi_{i}={\left(1-\lambda_{i}\right)}^{2}$$
(8)$$K_{d,i}=1-2.3\lambda_{i}+1.154\lambda_{i}^{2}+0.224\lambda_{i}^{3}$$
(9)$$K_{c,i}=\left(2-\phi_{i}\right)\left(1+0.054\lambda_{i}-0.988\lambda_{i}^{2}+0.441\lambda_{i}^{3}\right)$$
(10)$$\frac{dc_{i}}{dx}=\frac{V_{s}}{K_{d,i}D_{i}}\left(K_{c,i}c_{i}-C_{p,i}\right)-\frac{F}{RT}z_{i}c_{i}\frac{d\psi }{dx}$$
(11)$$\frac{d\psi }{dx}=\frac{\sum_{i=1}^{NoComp}\left(\frac{z_{i}V_{s}}{D_{i}}\left(K_{c,i}c_{i}-C_{p,i}\right)\right)}{\frac{F}{RT}\sum_{i=1}^{NoComp}\left(z_{i}^{2}c_{i}\right)}$$
(12)$$\sum_{i=1}^{NoComp}c_{m1i}z_{i}=-X_{d}$$
(13)$$\sum_{i=1}^{NoComp}C_{p,i}z_{i}=0$$
(14)$$c_{m,i}=C_{f,i}\phi_{i}\exp\left(-\frac{z_{i}F}{RT}\psi_{D}\right)$$
(15)$$R_{i}=1-\frac{C_{p,i}}{C_{f,i}}$$

Boundary conditions:(16)$$c_{i}(0^{+})=c_{m1,i}$$
(17)$$c_{i}(\Delta x^{-})=C_{p,i}$$

All model variables and model equations were reported and described in Table 1 and Table 2, respectively. The degree of freedom (DOF) of the presented model is equal to 8+8·NC, where NC stands for number of compounds present in the mixture. The values of diffusion coefficients and sizes of ions were reported in Table 3, while the permeate fluxes obtained for each variant were around 8.06 × 10^−5^ m/s.

Using the mathematical model presented above, it was possible to obtain the total volume membrane charge density (*X_d_*). The parameter estimations of total volumetric membrane charge density were performed for the sets of variants listed in Table 4. The parameter estimation computations were conducted using the gPROMS software. Parameter estimation in gPROMS is based on the maximum likelihood formulation, which provides a simultaneous estimation of parameters in the physical model of the process [70]. Assuming independent, normally distributed measurement errors—with zero means and standard deviations—the maximum likelihood goal can be achieved through the objective function presented in Equation (18) [70]. In the cases discussed in this study, the parameter estimation problems gave the following values of parameters following Equation (18): *NE* = 3, *NV* = 1, *NM* = 1, *N* = 3.
(18)$$\Phi =\frac{N}{2}\ln\left(2\pi \right)+\frac{1}{2}\min X_{d}\left\{\sum_{i=1}^{NE}\sum_{j=1}^{NV_{i}}\sum_{k=1}^{NM_{ji}}\left[\ln\left(\sigma_{ijk}^{2}\right)+\frac{{\left(c_{ijk,mes}-c_{ijk}\right)}^{2}}{\sigma_{ijk}^{2}}\right]\right\}$$
where: Φ—set of model parameters to be estimated, *N*—total number of measurements taken during all the experiments, *NE*—number of experiments performed, *NV_i_*—number of variables measured in the *i*-th experiment, *NM_ij_*—number of measurements of the *j*-th variable in the *i*-th experiment, *σ*^2^*_ijk_*—variance of the *k*-th measurement of variable *j* in experiment *i*, c*_ijk_*—*k*-th measured value of variable *j* in experiment *i*, c*_ijk,mes_*—*k*-th predicted value of variable *j* in experiment *i*.

## 3. Results and Discussion

### 3.1. Comparison of Standard and Detailed DSPM Nanofiltration Modelling

The parameter estimations were performed with the use of the above presented mathematical model. The correctness of the presented modelling approach can be acknowledged through a comparison of the estimated *X_d_* values for the standard approach—when only ions coming from the salt are considered—with *X_d_* values estimated with use of the detailed modelling, which takes into account each existing ion in the feed solution. Such a comparison was presented for aqueous solutions of Cu(NO_3_)_2_. The obtained results are presented in Figure 3. In the standard approach, *X_d_* ranged between +282.79 and +982.87 [mol/m^3^], with a pH increase from 2 to 9, while in the detailed approach, it changed from −37.57 to +890.62 [mol/m^3^]. It is important to notice that the detailed DSPM model revealed a shift of total volume membrane charge to negative values between pH values equal to 4.8 and 6, which can be related to the presence of a specific isoelectric point of the separated mixture. It can be concluded that the presence of ions originating from the solution regulating the pH influenced the membrane *X_d_* values, which is clearly visible in the detailed model. This is related to the fact that all ions and molecules present in the system may interact with each other and, therefore, influence the charge present on and in the membrane. Therefore, all results presented hereafter were obtained with use of the detailed described DSPM model.

### 3.2. Estimated Values of Total Volume Membrane Charge Density

The total volume charge densities of the ceramic TiO_2_ membrane as a function of pH for all the experimentally investigated solutions of asymmetric salts (namely: Cu(NO_3_)_2_, Pb(NO_3_)_2_, Fe(NO_3_)_3_, Zn(NO_3_)_2_), and the influence of pH on the retention of heavy metals, are presented in Figure 4, Figure 5, Figure 6 and Figure 7. For all ions, the trends of the retention curves were the same as the charge density curves in terms of their qualitative manner. All *R = f*(pH) and *X_d_ = f*(pH) curves possess the S-shape, with the inflexion in the range of pH between 4.9 and 6.0. In the case of asymmetric salt, Labbez et al. [62] have already shown that the dependency of the retention as a function of pH is described by the S-shaped curve. The values of retention rates obtained experimentally and by means of the detailed DSPM model were identical, and therefore, in this work, there is no difference in plotting experimental or calculated retentions.

In general, the possible mechanisms for the separation of electrolytes are sieving, electrostatic interactions between the membrane and the ions or between the ions mutually, differences in diffusivity and solubility, or a combination of all those listed [19,71]. A high retention for multivalent ions is frequently combined with a moderate retention for monovalent ions. In our study, the pore size of the membrane was large enough to demonstrate that salt retention is only affected by size effects to very little extent. Taking into account the difference between the membrane cut-off (which is equal to 450 Da) and the studied ion radii—which, e.g., for the Pb^2+^ ion (the largest of the investigated ions) is equal to 11.9·10^−11^ m—the steric effect may not justify the obtained ions retentions. For all experiments, the highest retention was achieved for Cu^2+^. At a pH equal to 9, retention reached values above 97%. For the Fe^3+^ and Zn^2+^ ions, the highest degrees of retention rates was also achieved for a pH = 9, but the values were much lower and equal to 80.3% and 58.8%, respectively. Whereas for the Pb^2+^ ion, the highest retention was achieved for pH = 6.9 which was 90.2%. Such values of retention could be related to the differences in diffusivities and electrostatic interactions between ions and membrane. The maximum retention for Cu^2+^ may have resulted from the lowest values of diffusion coefficient of all ions and the minimum retention of Zn^2+^ from the highest diffusion coefficient of all ions (compare with Table 3).

For the estimated values of the total volume charge density for each set of ions present in the aqueous solutions, the membrane becomes different in terms of individual charge, or—in other words—the apparent charge densities on and in the membrane are significantly different. That dependence is associated with the nature of the electrolyte in the system and with the specific adsorption on the membrane surface and pore walls. For solutions containing Cu^2+^ ions (1st variant in Table 4), the *X_d_* varied with pH changes in the range from 37.6 to 890.6 mol/m^3^; for the 2nd variant (Table 4), for solutions containing Fe^3+^ ions from −120.9 to −37.0 mol/m^3^; for the 3rd variant (Table 4), containing Zn^2+^ ions from −289.4 to −150.9 mol/m^3^ and for last variant, which contained Pb^2+^ ions, from -245.0 to 105.6 mol/m^3^. At first glance, the variation of the *X_d_* sign is surprising, especially due to the fact that all of the investigated heavy metals were in ion forms. It should also be noted that NO_3_^−^ ions were present in all the listed variants. They were present because the investigated cations were introduced into the solution in the form of nitric (V) salt. Moreover, Na^+^, OH^-^ and Cl^−^, H^+^ ions were present in the aqueous solution, which originate from sodium base and hydrochloric acid, respectively—used for the regulation of pH.

The obtained inflections of the membrane charge density curves for all ions were confirmed; in each system, the minimal value of total volume membrane charge density was in the range of pH 4.5–6.0, which corresponds to the IEP of the studied membrane. For the Cu^2+^, Fe^3+^, and Pb^2+^ ions, the minimal values of *X_d_* were for pH = 4.6 and for Zn^2+^ for pH = 6. The type of membrane material used for the active layer influences the membrane structure, and thereby affects the membrane separation ability, but also has an influence on membrane surface charge, which depends on the material isoelectric point value. The membrane possessed a positive charge during the filtration of separated solutions with pH lower than the IEP value, whereas during the filtration of solutions with a pH higher than IEP, the membrane possessed a negative charge. Therefore, the obtained trend of total volume membrane charge densities is correct. For example, when Cu^2+^ ions are present in the system at a pH below IEP, the Cu^2+^ ions are repelled and the anions present in the feed solution are bound to the membrane, so that the overall stable charge on the membrane during that separation is negative and the retention level is lower due to the formed negative layer which attracts Cu^2+^ cations. In cases when the pH of the feed is higher than that of the IEP, the Cu^2+^ ions are attracted, and thus retention increases and the change in overall membrane charge *X_d_* might reflect the partial surface adsorption of cations. Such behaviour of the membrane at different pH values is explained by the amphoteric behaviour of the TiO_2_ active membrane layer reported by Van Gestel et al. [27], which is schematically visualised in Figure 8. Unfortunately, the relation between IEP and the inflection point for the obtained curves for all investigated cases—when di(tri)-monovalent salts were studied—does not work properly for mono-monovalent salts. Van Gestel et al. [27] studied zeta potential measurements as a function of pH for mono-monovalent and mono-divalent salts (Na_2_SO_4_, CaCl_2_). They concluded that mono-monovalent salts can be considered as indifferent electrolytes for the NF membrane, and that the inherent charge is due to the protonation and dissociation of surface hydroxyl groups (IEP = 6). Whereas for mono-divalent salts, that trend was totally different. The sign of the zeta potential is altered with the type of salts and salt concentrations. Those phenomena were explained by the selective adsorption of cations or anions. Depending on the forms of –Ti–OH surface groups, ions are able to form complexes. Increasing values of membrane charge densities may be caused by the selective adsorption and additional ions adsorption; the first stage is complexation and the next is the adsorption of additional ions. Moreover, some ions may be adsorbed by the pore wall and influence the membrane charge, as suggested by Takagi et al. [72]. 

It is postulated that the total volume membrane charge density is determined by the sum of the fixed membrane charge density and the number of adsorbed ions. The possible mechanism for the formation of the membrane charge assumes that the ions are partitioned from the bulk solution into the membrane pore under the influence of the Donnan potential. Among the partitioned ions in the membrane pores, either cations or anions are adsorbed selectively by the pore wall. Next, the adsorbed ions are bound on the pore wall and give the electric charge to the membrane. In our opinion, the electric charge given to the membrane, which includes all these phenomena, can be seen as the total volume membrane charge density, as presented in Figure 8. In view of this, the values of total volume membrane charge density *X_d_* will always be different depending on the type of solute (electrolyte) which is subjected to the NF process, and hence, on the ion types and the pH values as well. Such dependence was obtained for the investigated solutions. For each studied solution, the *X_d_* values in pH range varied from 2.0 to 9.0 are different. Therefore, it can be assumed that the mechanism of selective ion adsorption acts according to the membrane sign, which is positive at a low pH and negative at a high pH; cations or anions are adsorbed on the membrane (see Figure 8), changing the values and, in two cases, the charge of *X_d_,* which is also visible in Figure 4, Figure 5, Figure 6 and Figure 7. Figure 8 shows the possible explanations of the transport of copper ions below and above IEP; however, it also should be considered as a general explanation of ion transport, whether transports of di- or tri-valent ions are studied.

Normally, the membrane became more negative at a higher pH of the feed. It needs to be highlighted that such a trend exists for monovalent salts—for example, NaCl. In this work, asymmetrical salts were considered and the observed trends were similar to those presented by Mazzoni et al. [73]. Additionally, the *X_d_* values stated in this work are values of membrane charge density after nanofiltration process stabilization, i.e., in steady state operation. The membrane active layer functional groups (TiO_2_) take forms which depend on the pH of the feed solution contacting the membrane surface, therefore obtaining the adequate surface charge. With the advent of such charge, adsorption and charge exclusion occur, leading to stable separation and reaching the estimated *X_d_* values.

In an acidic environment, metals occur in the form of free ions, and the absence of soluble charged metal hydroxides render the formation of an additional active separation layer on the membrane surface impossible. At low pH values, retentions are always lower than when pH increases. As the pH of the environment increases, so too does the amount of soluble metal hydroxides. Due to electrostatic effect of the separated mixture, i.e., metal hydroxides–membrane interaction, an active filtration layer can form on the membrane surface, and the retention rate increases, for Cu^2+^ from 72% (pH = 2) to 97.7% (pH = 9), for Fe^3+^ from 70.2% (pH = 2) to 80.3% (pH = 9), for Pb^2+^ from 56.8% (pH = 2) to 86.20% (pH = 9), and for Zn^2+^ from 36.1% (pH = 2) to 58.5% (pH = 9). The formation of that layer results in an increase in the density of positive charge in the membrane, which causes the cation retention to increase for all of the investigated experimental variants, as is also presented in [60]; the values of total volume membrane charge density for each variant increase, which is also presented in Figure 4, Figure 5, Figure 6 and Figure 7. 

Divalent ions have an important effect on the surface charge—divalent cation adsorption on the membrane surface reduces its negative charge. On the other hand, when both divalent cations and anions are present in the solution, the effect of the divalent anion is opposite to the effect of the divalent cation [35]. Therefore, the obtained total volume membrane charge densities can be related to the apparent interactions between ions present in the mixtures. These phenomena can explain the observed different values of *X_d_* for different ions, because, as mentioned above, for each variant, all ions present in each system were taken into account. For example, for the Cu^2+^ variant in the system, ions such as NO_3_^−^, OH^−^, Na^+^, H^+^ were also considered. Therefore, besides Cu^2+^ and membrane interactions, all various phenomena associated with those ion–ions interactions (selective absorption, Donnan partitioning) occur, which significantly influence the total volume membrane charge density values which are inherently included. Additionally, changes in the additional ions ratio in the systems influence the pH values of the feeds.

Generally, when the membrane makes contact with the aqueous electrolyte solution, it takes the electric charge according to a few possible mechanisms: functional group dissociation, the adsorption of ions from solution, and the adsorption of polyelectrolytes, ionic surfactant or charged nanoparticles. Such charge has an influence on ions distribution in the solution, in view of the electroneutrality requirements of the separated system [74]. This charging mechanism can occur on the exterior membrane surface and in the interior pore surface, due to the distribution of ions in the solution to maintain the electroneutrality of the system [75]. The membrane has the internal and surface charge density. Surface charge may be assigned to constant membrane charge (intrinsic), which is generated when the membrane is soaked in the electrolyte. This is caused in view of the acid/base dissociation or ionization of other functional groups, or ions adsorption on the membrane surface from the solution. Therefore, in this study, the overall membrane charge was considered, which presents the total volume membrane charge density created during the NF separation.

In order to enable a comparison of the obtained data with the literature data, the effective membrane charge density was rearranged to the surface charge density according to Equation (19), with the assumption that membrane surface charge is uniformly distributed on the entire intergranular volume between cylindrical pores [39]:(19)$$\sigma =\frac{X_{d}r_{p}F}{2}$$
where *σ* is the surface charge density [C/m^2^], *r_p_* is the pore radius [m], and *F* is the Faraday constant [C/mol]. The values of the total volume membrane charge densities after conversion to surface charge densities *σ* [C/m^2^] are presented in Table 5. These values are in good qualitative agreement with the values presented in [76].

### 3.3. Determination of Corellation of the Total Volume Membrane Charge Density 

In order to determine the correlation which would provide at least limited re-use of the obtained estimation results of the presented modelling, the correlations of the estimated total volume membrane charge densities were obtained. In the trial-and-error search of the feasible form of a correlation relating *X_d_* and pH, including the Newton’s and Lagrange’s interpolating polynomial methods, Equation (20) was finally proposed: (20)$${\hat{X}}_{d,i}(pH)=\frac{a{\left(pH\right)}^{2}+b\left(pH\right)+c}{d{\left(pH\right)}^{2}+e\left(pH\right)+f}$$
where *a*, *b*, *c*, *d*, *e*, *f* are the coefficients, the values of which are presented in Table 6 as first set of parameters.

The parameters of correlation were regressed with use of the least squares method, and the so-defined objective function reached values between 0.50 for Pb^2+^ and 611.44 for Zn^2+^. The presented form of Equation (20) gives the first view of how function *X_d_*
*= f*(pH) might be shaped, and through which values it can progress.

In this study, the measure of model compatibility with empirical data was based on the variance of random component method. The starting point was model residuals. The assessment of the random component variance, the so-called remainder variance, is expressed by Equation (21):(21)$$S_{e}^{2}=\frac{{\sum_{i=1}^{n}\left(X_{d,i}(\mathrm{pH})-{\hat{X}}_{d,i}(\mathrm{pH})\right)}^{2}}{n-m-1}$$
where *X_d,i_*(pH) is the total volume membrane charge density determined experimentally [mol/m^3^], ${\hat{X}}_{d,i}(\mathrm{pH})$ is the total volume membrane charge density calculated with regression model [mol/m^3^], *n* is the number of observations, and *m* is the number of estimated model parameters.

The root of the remainder variance is the standard deviation of the residues *S_e_* (also known as the estimation standard error). This value indicates the average difference between the observed values of the explanatory variable and theoretical values. As seen in Figure 9a, the obtained correlations converge well with the computationally obtained results of *X_d_*. In Figure 9, the horizontal thin lines mark the range of IEP, whereas the horizontal bold line marks the value of the IEP of the TiO_2_ membrane. As mentioned earlier, the obtained inflection of the membrane charge density curves for all ions is confirmed, and in each system the minimal value of total volume membrane charge density was in the range of pH 4.5–6.0. The shapes of the obtained correlation functions are in good agreement, and the inflection points of each ion are generally close to the limits of IEP, except for solutions containing Fe^3+^.

After the analysis of the first set of parameters reported in Table 6, it was proposed to unify the parameters present in the denominator of Equation (19) for divalent cations and perform the parameter optimization. The results of those optimizations are reported as the second set of parameters in Table 6. Although the second set of parameters exhibit higher values of *S_e_* in comparison to the first set, they are still in good quantitative agreement (see Figure 9b).

## 4. Conclusions

The main aim of the computer-aided estimations and simulations performed was to estimate the total volume membrane charge density using the Donnan–Steric partitioning model, derived from the extended Nernst–Planck equation with the Donnan partitioning assumption, which finally resulted in a relation between the total membrane charge density and the pH of separated solutions. The obtained total volume membrane charge densities reflect the experimental values of ion retention very well. The values of such charge densities of the membrane are very important for the explanation of the possible mechanisms of ions transport across the membrane, which regulate the value of the solute retention and influence the electrostatic repulsion between the ions and the membrane.

The value of total volume membrane charge density is influenced by several factors. One of the key factors is the type of the solution, which is directly related to the valance of present ions. However, the determination of the *X_d_* value requires an experimental investigation for each NF system. 

The obtained results of total volumetric membrane charge densities confirm the amphoteric behavior of ceramic Al_2_O_3_/TiO_2_ NF membranes. The *X_d_* values change with the increase in the pH of feeds. Initially, at low pH values, when the membrane is positively charged, all types of ions obtained negative values for the total volume membrane charge densities. This is associated with the adsorption of ${\mathrm{NO}}_{3}^{-}$ ions due to electrostatic attraction. Next, when the pH values began to increase, the *X_d_* also changed. For all types of ions, the *X_d_* increased, but in a different manner. For solutions containing Cu^2+^ and Pb^2+^ ions, their *X_d_* values increase from negative to positive values, and for Fe^3+^ and Zn^2+^, increases were also observed, but the values nevertheless remained negative. For Cu^2+^ from pH 2 to pH 9, the total volume membrane charge density changed dramatically, by approx. 738%, and for Pb^2+^, the change was the smallest, at approx. 292%. The fact that the values for each ion variants increase is undoubtedly associated with ions–membrane interactions, precisely the with electrostatic attraction of cations. Our studies provide an interesting unexplained observation, that for Fe^3+^ and Zn^2+^, the values of the total membrane charge densities are negative in the whole range of pH. Additionally, lower retention rates were achieved for those two ions in comparison to solutions that contained Cu^2+^ or Pb^2+^. This effect may be caused by ions complexation and a strong interaction between ions and ions present in the feeds. It is also worth noting that, if the total membrane charge density is strongly positive, the retention rate is significantly better than when the membrane charge is negative. Equation (20) in the presented form may allow for the easier prediction of the retention rate of the studied solutions. Such a correlation allows its use in process simulations, i.e., by knowing values of pH of aqueous solution of specific ion, the values of *X_d_* in the considered system can be computed, therefore leading to the calculation of metal ion retention.

The total volume membrane charge density is very difficult to determine without performing any experiments. Based on the obtained results, it can be seen that even for groups of cations with the same valence, the *X_d_* has significantly different values. It is also difficult to assess which mechanisms play a key role in shaping the membrane charge, whether Donnan partitioning, selective adsorption, electrostatic interaction, diffusion difference, or other, as-yet-unexplored phenomena. Therefore, mathematical modelling with closely associated experimental studies must be further carried out.

The additional and highly important output from this study is that the presented method based on the model and parameter estimation allows for a first view of the separation mechanism without time consuming studies of intrinsic charge. Therefore, knowledge regarding the total volume membrane charge density helps in the systematic investigation of the influence of membrane charge on the behaviour of salts, even if it is computation-based and obtained through the parameter estimation of rejection experiments. The value of total volume membrane charge density in the presence of a few ions cannot be interpreted in the same way when a single pair of ions are present in the system. In other words, interactions between all ions in the investigated systems should be taken into account. However, the authors consider that, in the future, it should be possible to define the relation between specific ions and total volume membrane charge density through some mixing rules, which would allow for the real predictive modelling of NF separations based on a few retention experiments.

## Figures and Tables

**Figure 1 membranes-10-00235-f001:**
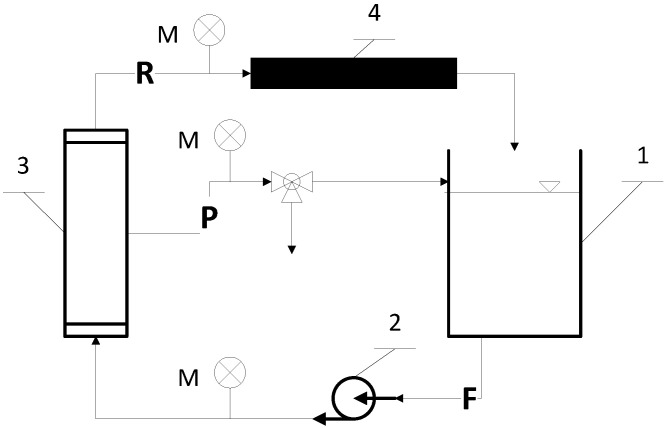
Schema of the laboratory plant: 1—feed/retentate tank; 2—pump; 3—membrane module; 4—radiator; F—feed; R—retentate; P—permeate; M—manometer.

**Figure 2 membranes-10-00235-f002:**
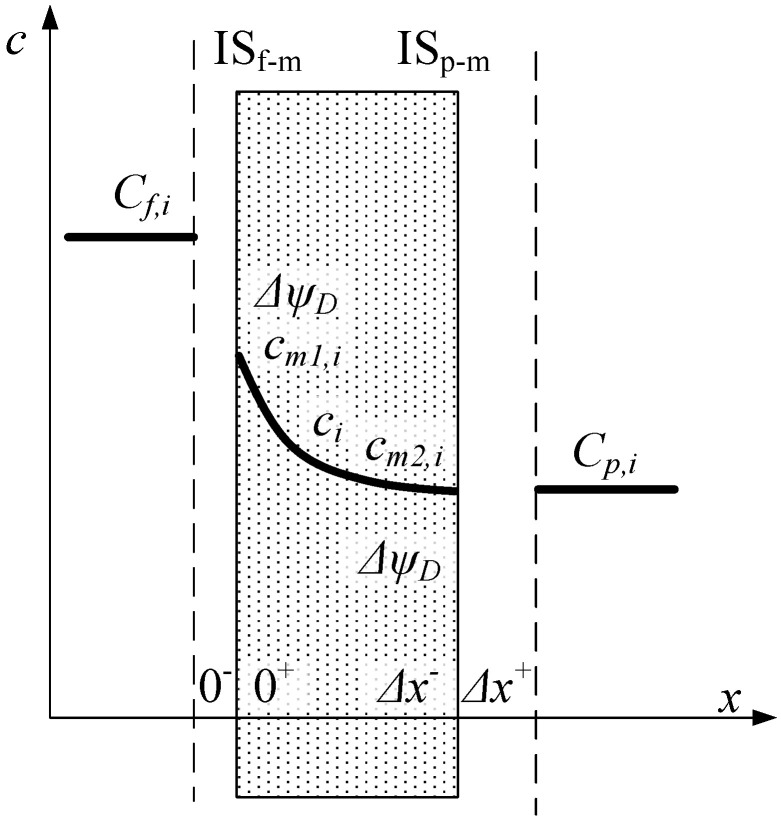
Concentration profiles of ions in the membrane active layer and external solutions, considering the Donnan potentials, where *C_f,i_* and *C_p,i_* are the concentrations of individual ion in the feed and permeate solution, respectively; Δ*x* is the membrane active layer thickness; *c_m_*_1*,i*_ and *c_m_*_2*,i*_ are the concentrations of the individual ion at both the feed and permeate boundaries, respectively; *c_i_* is the concentration of individual ion along the pores. *IS_f-m_* and *IS_p-m_* represent the interfacial surfaces of feed–membrane and permeate–membrane, respectively.

**Figure 3 membranes-10-00235-f003:**
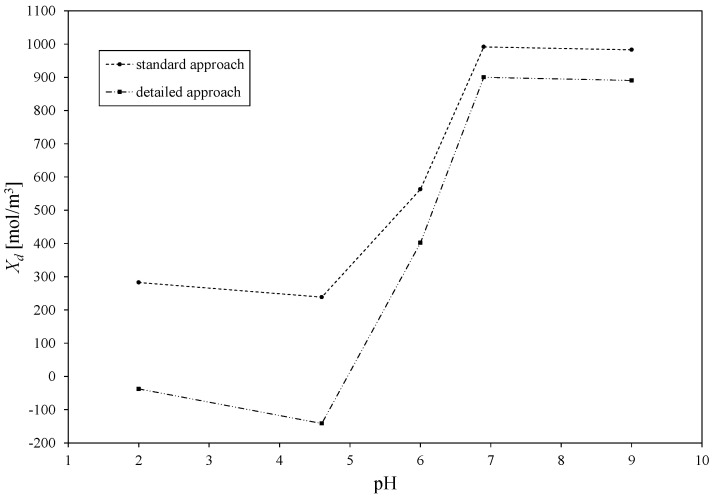
Comparison of estimated total volume membrane charge densities *X_d_* for standard and proposed modelling approach with detailed described DSPM model, for aqueous solution of Cu(NO_3_)_2_.

**Figure 4 membranes-10-00235-f004:**
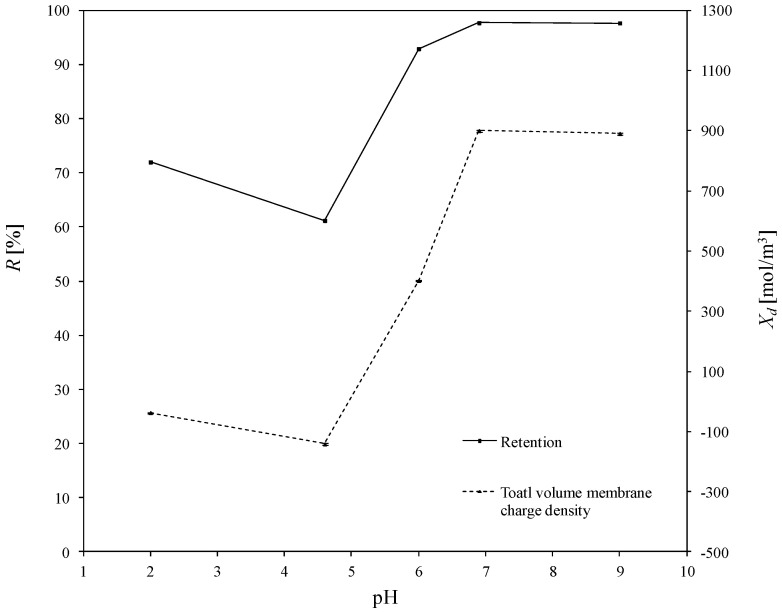
The influence of pH on the total volume charge density in aqueous solution of Cu(NO_3_)_2_ and the influence of pH on the Cu^2+^ retention. Retention values obtained experiments.

**Figure 5 membranes-10-00235-f005:**
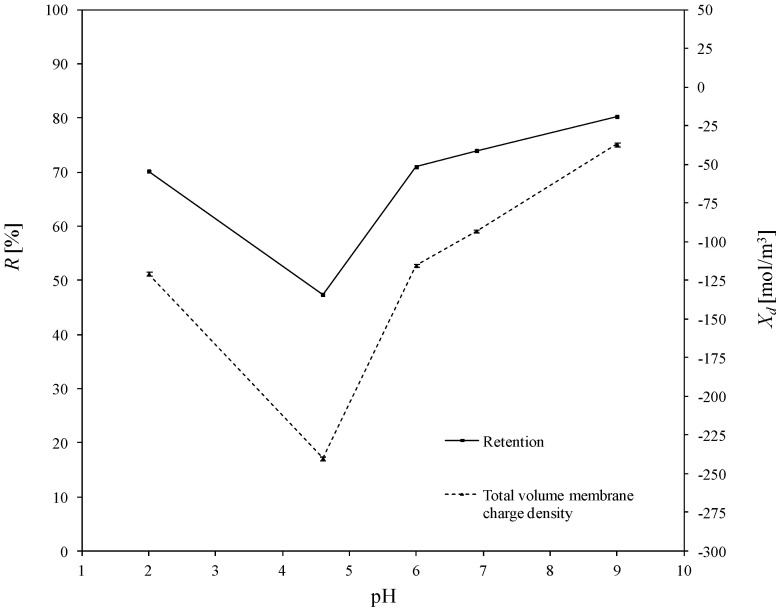
The influence of pH on the total volume charge density in aqueous solution of Fe(NO_3_)_3_ and the influence of pH on the (Fe^3+^) retention. Retention values obtained experiments.

**Figure 6 membranes-10-00235-f006:**
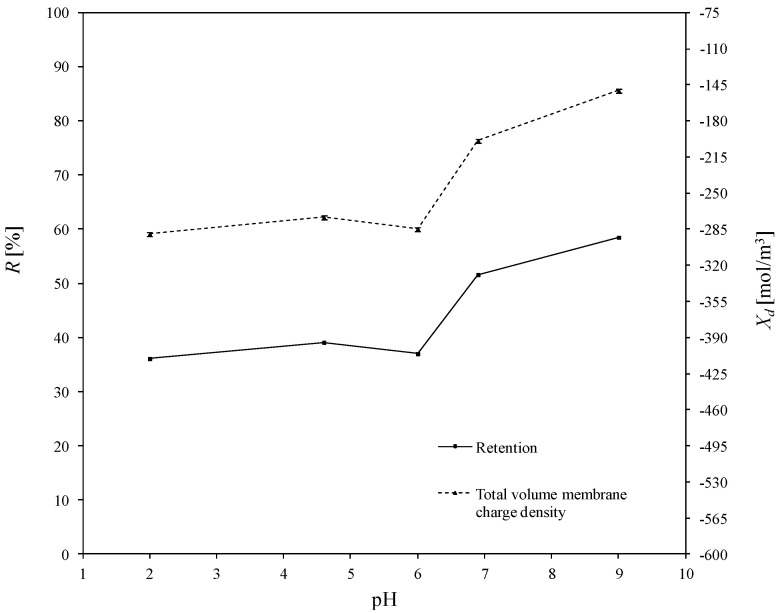
The influence of pH on the total volume charge density in aqueous solution of Zn(NO_3_)_2_ and the influence of pH on the (Zn^2+^) retention. Retention values obtained experiments.

**Figure 7 membranes-10-00235-f007:**
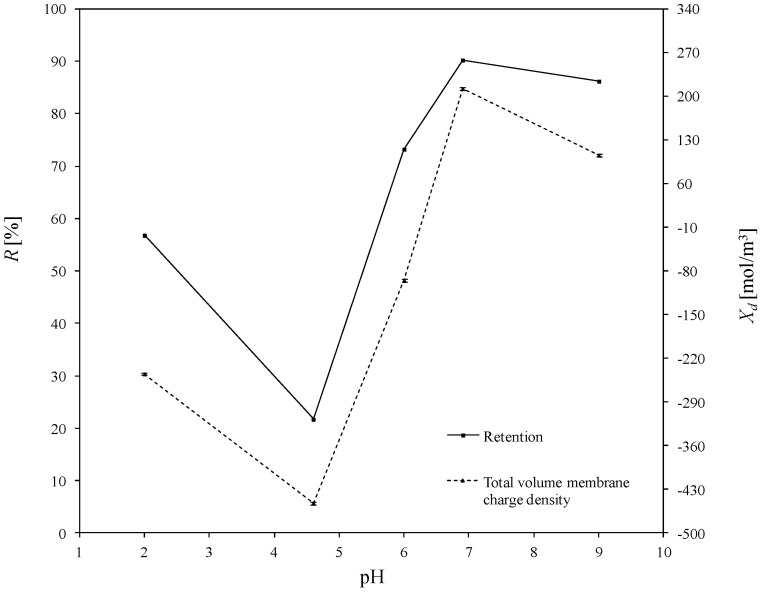
The influence of pH on the total volume charge density in aqueous solution of Pb(NO_3_)_2_ and the influence of pH on the (Pb^2+^) retention. Retention values obtained experiments.

**Figure 8 membranes-10-00235-f008:**
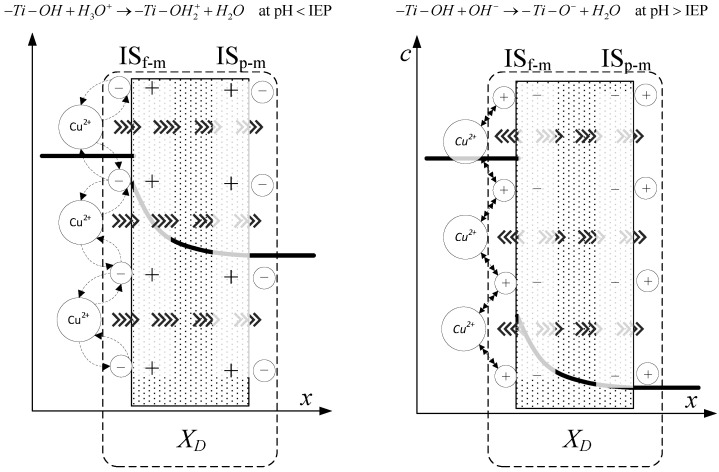
Schematic representation of amphoteric behavior of TiO_2_/Al_2_O_3_ active layer of membrane during separation of aqueous solution of Cu(NO_3_)_2_.

**Figure 9 membranes-10-00235-f009:**
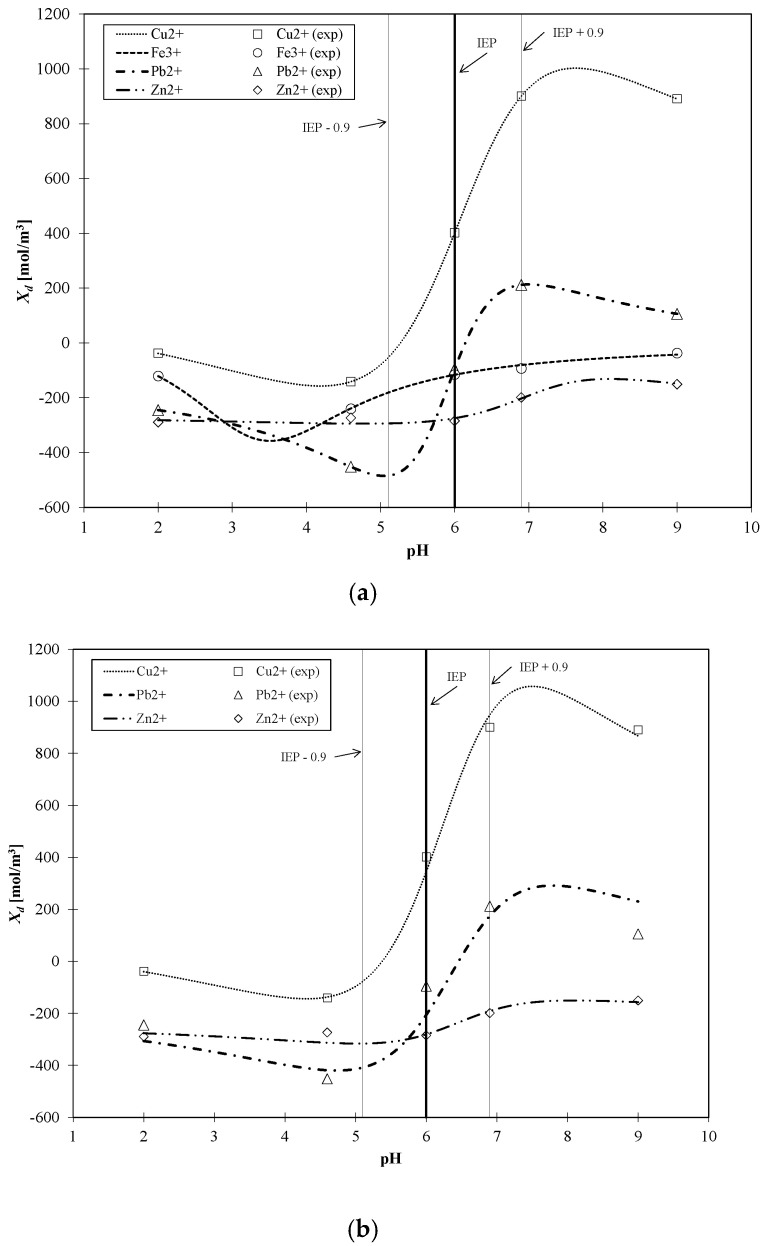
Performance of correlation Equation(20), with parameters listed in Table 6, with respect to estimated *X_d_* values. (**a**) first set of parameters from Table 6; (**b**) second set of parameters from Table 6.

**Table 1 membranes-10-00235-t001:** Variables in Donnan–Steric partitioning model (DSPM) (NC—number of components).

**Differential Variables**		**Number**
Concentration of ion in the membrane [mol/m^3^]	*c_i_*	*NC*
**Algebraic and Implicit Variables**		**Number**
Potential gradient inside membrane pore [V]	*ψ*	1
Ratio of solute to pore radius [-]	*λ* *_i_*	*NC*
Steric term [-]	*φ* *_i_*	*NC*
Hindrance factor for diffusion [-]	*K_d,i_*	*NC*
Hindrance factor for convection [-]	*K_c,i_*	*NC*
Ion concentration in the permeate [mol/m^3^]	*C_p.i_*	*NC*
Retention coefficient [-]	*R_i_*	*NC*
Solvent velocity [m^3^/m^2^/s]	*V_s_*	1
Donnan potential [V]	*ψ* *_D_*	1
Osmotic pressure difference [Pa]	Δ*π*	1
Osmotic pressure on the feed side [Pa]	*π* *_feed_*	1
Osmotic pressure on the permeate side [Pa]	*π* *_permeate_*	1
**Parameters and Known Variable**		**Number**
Effective membrane charge density [mol/m^3^]	*X_d_*	1
Molar fraction on the feed side [mol/mol]	*x_f,i_*	*NC*
Molar fraction on the permeate side [mol/mol]	*x_p,i_*	*NC*
Pore radii [m]	*r_p_*	1
Ion radii [m]	*r_s,i_*	*NC*
Transmembrane pressure [Pa]	Δ*P*	1
Ideal gas constant [J/(mol⋅K]	*R*	1
Faraday constant [C/mol]	*F*	1
Temperature [K]	*T*	1
Solvent viscosity [Pa⋅s]	*η* *_s_*	1
Thickness of membrane active layer [m]	Δ*x*	1
Molar volume of water [m^3^/mol]	*Ṽ_w_*	1
Diffusion coefficient of ion [m^2^/s]	*D_i_*	*NC*
Charge of individual ion [e]	*z_i_*	*NC*
Ion concentration in the feed [mol/m^3^]	*C_f,i_*	*NC*
Ion concentration in the membrane in the surface directly contacting with the feed [mol/m^3^]	*c_m_* _1*,i*_	*NC*
Ion concentration in the membrane in the surface directly contacting with the permeate [mol/m^3^]	*c_m_* _2*,i*_	*NC*
**Total number of variables: ** 15+15·NC		

**Table 2 membranes-10-00235-t002:** List of equations in the DSPM model (NC—number of components).

Equation Description	Equations	Number of Equations
Solvent velocity based on Hagen–Poiseuille-type relationship	(2)	1
Osmotic pressure difference across the membrane	(3)	1
Osmotic pressure at the feed side	(4)	1
Osmotic pressure at the permeate side	(5)	1
Ratio of the solute radii to the pore radii	(6)	NC
Steric partitioning coefficient	(7)	NC
Hindrance factor for diffusion	(8)	NC
Hindrance factor for convection	(9)	NC
Concentration gradient inside the membrane pore	(10)	NC
Potential gradient inside the membrane pore	(11)	1
Electroneutrality conditions in the membrane	(12)	1
Electroneutrality conditions in the permeate	(13)	1
Donnan–Steric partitioning	(14)	NC
Retention coefficient	(15)	NC
**Total Number of Equations:** 7 + 7·NC		

**Table 3 membranes-10-00235-t003:** Values of diffusion coefficients and size of ions.

Ion	Diffusion Coefficient, *D_i_* [m^2^/s]	Size of Ion/Molecule, *r_s,i_* [m]
Cu^2+^	1.24 × 10^−9^ [63]	7.7 × 10^−11^ [64]
Fe^3+^	7.19 × 10^−9^ [65]	6.0 × 10^−11^ [64]
Zn^2+^	5.18 × 10^−8^ [65]	7.4 × 10^−11^ [64]
Pb^2+^	8.45 × 10^−9^ [65]	11.9 × 10^−11^ [64]
${\mathrm{NO}}_{3}^{-}$	1.25 × 10^−9^ [66]	1.79 × 10^−10^ [67]
H^+^	4.50 × 10^−9^ [68]	0.9 × 10^−9^ [64]
Na^+^	1.33 × 10^−9^ [69]	0.1 × 10^−9^ [64]
OH^−^	5.27 × 10^−9^ [69]	1.33 × 10^−10^ [67]

**Table 4 membranes-10-00235-t004:** Variants of parameters estimation.

Variant	Heavy Metal Ion *	Heavy Metal Ion Concentration [mol/m^3^]	pH				
			1-Set	2-Set	3-Set	4-Set	5-Set
1.	Cu^2+^	7.87 × 10^−3^	2.0	4.6	6.0	6.9	9.0
2.	Fe^3+^	8.95 × 10^−3^	2.0	4.6	6.0	6.9	9.0
3.	Zn^2+^	7.69 × 10^−3^	2.0	4.6	6.0	6.9	9.0
4.	Pb^2+^	2.41 × 10^−3^	2.0	4.6	6.0	6.9	9.0

* All ions were introduced as nitric salts of specific heavy metal.

**Table 5 membranes-10-00235-t005:** Values of effective charge density after conversion to surface charge density *σ*.

pH	Cu^2+^	Fe^3+^	Zn^2+^	Pb^2+^
2.0	−0.00082	−0.00262	−0.00628	−0.00532
4.6	−0.00307	−0.00521	−0.00594	−0.00981
6.0	0.00873	−0.00250	−0.00618	−0.00207
6.9	0.01954	−0.00202	−0.00432	0.00461
9.0	0.01933	−0.00080	−0.00328	0.00229

**Table 6 membranes-10-00235-t006:** Values of parameters in Equation (20).

Set of Parameters	Parameters	Fe^3+^	Cu^2+^	Zn^2+^	Pb^2+^
First set of parameters	a	4.44	342.07	−76.42	−247.56
	b	−199.89	−2251.68	1557.69	3754.86
	c	−40.6	2369.91	−6684.97	−14,589.5
	d	1	1	1	1
	e	−6.55	−12.30	−11.72	−14.27
	f	12.59	40.73	35.28	53.18
	*S_e_*	8.03	0.549	14.276	0.408
Second set of parameters	a	-	277.57	−92.68	−223.51
	b	-	−1795.30	2263.50	3104.14
	c	-	1630.37	−10,754.65	−11,272.98
	d	-	*1.00*	*1.00*	*1.00*
	e	-	*−12.77*	*−12.77*	*−12.77*
	f	-	*43.06*	*43.06*	*43.06*
	*S_e_*	-	*3.19*	*106.74*	*24.85*

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
