# Peer review of "Effect of pH on Total Volume Membrane Charge Density in the Nanofiltration of Aqueous Solutions of Nitrate Salts of Heavy Metals"

_membranes, 2020, doi:10.3390/membranes10090235_

Round 1

Reviewer 1 Report

This contribution in the field of nanofiltration relates the pH with an estimated membrane charge for single solutions of nitrate salts of heavy metals.   

The article has some interest for the specific interaction of this type of solutions and pH with the membrane.

However, it appears that some of the conclusions are over-generalized and an effort should be made in the text to distinguish what is general result and what it is specific of the solution.

1) Title

The title is too general it gives the impression that the effect of pH is studied for solutions containing different heavy metals. In my opinion, it must be more specific.

You can change in the same way you say in Line 100:  “… single aqueous solutions of nitrate salts of heavy metals”

Besides: perhaps “effect” is better than “impact”.

2) Abstract:

There are some errors or inaccuracies.

@ -> at

Computation à fitting

Yielded in à yielded to

“Novel approach” is not accurate. This fiiting of Xd has been used in the past.

“Such as NaOH or HCl” à Not necessary

3) English:

There are a lot of spelling and grammar errors that should be corrected. Please check again. Here you have some examples:

Line 38:  mg/L is recommended instead of mg/l

Line 39: polluting heavy metals

Line 40: Do not use the plural in “ores” when used like an adjective:  In ore enrichment plants (This type of error is constantly repeated)

Line 54: the membrane processes more and more replace conventional techniques

Line 76: to use the streaming potential technique / to use streaming potential techniques

Line 92: in à at various pH

Line 160 and 163: revile or reveal?

Line 193: explained by the above

There are others spelling errors or inappropriate use of tenses after these lines. Please perform a thorough revision, use a grammar corrector and correct them.

4) Sentences that are not clear :

Lines 43 and 44: It can guess what you want to say

Line 102: and no participation was observed in all investigated systems

Line 205: especially that the all investigated ions are cations

Line 223: In case of separated solutions

Line 404: Therefore, knowledge of the theoretical, although it is experimental-based, total volume membrane charge density, or computation-based through parameter estimation, helps in systematic investigation on the influence of membrane charge on the salts behaviour.

5) Superfluous sentences to remove or change:

Line 56: One of the newer pressure methods for separation of liquid mixture is nanofiltration (NF)

(NF is been used for 50 years!)

“It should be noted, that dependencies of retentions from pH illustrated on the Figures 3-6 are values determined experimentally. Nevertheless, values of retention rates achieved by our detailed DSPM model are identical and therefore in this work there is no difference in plotting experimental either calculated retentions”. (State that are experimental values in the Figures)

Line 232: “reported also somewhere else [25]” (Suppress or use “reported by Gestel et al. [25])

6) References needed:

Lines 50, 51: appear to be cheaper or more effective than traditional ones (reference needed)

There are different sources for correlations of Kdi and Kci (8) and (9). Their origin should be referred.  

7) No inclusion of concentration polarization should be justified.

Line 98: Experiments have been performed at 4 m/s. This should be use to justify why you do not include concentration polarization in the model.

8) Lack of information of numerical methodology that must be included

The set of DSPM equations constitute an algebraic diiferential equation model but no information about the integration algorithm used or the solving strategy is provided.

The parameter estimation  has been performed in gPROMS, but in lines 143-149 or the Tables no information about the confidence of the intervals is provided (in this case for Xd). The objective function should be clearly stated as an equation.

Have retention values of the variants have been used as experimental data for fitting? In this case, they values should appear   

9) Correlation

There are 5 experiments and 6 coefficients are obtained. It is a pity because the inclusion of some additional pH would have allow to validate the generalization of expression (19). In fact, you have found a mathematical function shape that works well for the phenomenon studied.

Perhaps “Generalization and to generalize and re-use” should be suppressed and change to “Determination of correlation”

The order magnitude of the coefficients in Table 6 is very different due to the excess of coefficient. However you can easily solve it. Considerer expression (19) con coefficient d’ = 1. Therefore: a’ = a/d, b’ = b/d, c’ = c/d, e’ = e/d and f’ = f/d. The order of magnitude of the new coefficients  between ions would be more similar and perhaps you could add additional discussion. As you really use 5 experiments for 5 real coefficients the Se value is irrelevant and can be suppressed form Table 6.    

Reviewer 2 Report

Manuscript Number: membranes-900186

Title: Impact of pH on total volume membrane charge density in nanofiltration of aqueous solutions containing heavy metals

Authors: Agata Marecka-Migacz et al.

The present work explores the numerical studies of volumetric charge density of Zn, Cu, Fe, Pb and compares the retention coefficient of desired ions with the function of feed pH (from 2 – 9). The theme of this manuscript is not novel, but the results could have some importance in wastewater treatment, in particular for membrane mechanisms. Nevertheless, the result of retention coefficient of ions and the numerical investigations of volumetric charge density are not clear and it is matter of controversy.

The subject matter of the paper could be interesting, but the paper cannot be published in this present form. My specific comments are as below.

General comments:

            It is very difficult to understand this paper owing to the lack of information about all the presented results. Some reference are not related in results and discussion and the author need to recheck this section thoroughly.

Specific comments:

In Abstract,

The authors state that, ‘aqueous solution containing Zn, Cu, Fe or Pb…’ however, in results and discussion showed both Fe and Pb and need to re correct it. Further, it also stated that, ‘..a novel approach in modelling of nanofiltration process..’ There is a lot of works in the literature which deal with the investigation of volumetric charge density of membranes for given electrolytes

In Introduction,

In Page1, line 44, ‘…from industrial waste water….’ here wastewater is one word.

In Page2, line 55, Nanofiltration is well defined in pressure driven process of membrane technology and it’s not a new techniques.

In Page2, line 68, The author need to verify the sentence, because, the rejection mechanism of ions in charged membrane is depends on diffusion, convection and electrostatic or electro-migration.

In page2, line 76-78, ‘Thanks to this…’Is there any evidence, whether the membrane used to damage during zeta potential measurements?. Nowadays, lot of new techniques used to applicable for determination of zeta potential on the surface properties.

In page2, line 82, In general, transport model in nanofiltration system classified as two parts, namely irreversible thermodynamic model and mechanistic model.  Hence, I request the author to re correct the sentence as well reframe the introduction part.

Moreover, the authors need to update the recent research articles (last 10 year) as reference and remove unwanted references in results and discussion part.

This present study really must explain that provides this work compared to the literature.

In materials and methods

In Page3, line 101, whether the author prepared the stock solution of total salt concentration or specific ions concentration? for Zn as 500 µg/dm3 or Zn(NO3)2 as 500 µg/dm3 ..?Because in table 4, the author studied the specific ions concentration.

In equation 9, stated that, the solvent velocity or flux is proportion to diffusion, convection and electro-migration. However, the authors, didn’t mention about the diffusion and convection effects in this section.

As per equation 7, whether the author take the account of dielectric exclusion effects on this study..? if yes, how the author calculate the activity coefficient of solute radius as well as pore radius?.  Moreover, the dielectric exclusion playing a major role on electrolytes in the pore wall and it also used to determination of volumetric charge density of given electrolytes. 

In Table3, the author need to recheck the radius of Pb2+, because it mentioned as 11.9×10-11 m

 Results and discussion

In this section, the author need to rewrite the whole parts and define, what exactly required in results and discussion.  In addition, there is lot of unwanted references, which is not related on this study and most of reference need to move introduction part. Hence, the author need to verify each reference, whether it’s related to this work or not.

Specific comments

In page7, line 163, ‘…which is clearly reviled…’ whether it’s reviled or revealed..?

In page7, line 173, ‘… asymmetric salts (namely: Cu(NO3)2, Pb(NO3)2, Fe(NO3)3, Zn(NO3)2)..’ Again, the author need to clarify whether this study (in experimental and numerical), they used specific ions concentration or total ions concentration? Because its controversy between that sentences as well as table 4.

The whole write up of page 8, the author need to recheck their claims in manuscript. And its better to rewrite the whole page.

  • line 189, state that Pb radius is 1.9×10-11 m, however in table 3, it mentioned as 9×10-11 m.  Moreover, the pore radius (0.9nm) is higher than solute radius, hence the steric hindrance effect supposed to be negligible, what about the convective transport and other mechanism through the pores. Is the author check the Peclet (Pe) number.. ? Because when Pe >1 convective force is dominant, while for Pe<1 diffusive force is dominant and, Pe=1 both convective and diffusive forces are equal
  • Line 192, ‘…and equals to 80.3% and 58.8% respectively’ at what pH those values are obtained..?
  • Line 194, stated that, the difference between the rejection ratio may due to diffusivity of given solutes. It’s quite controversy, what about electrostatic interaction between the specific ions and surface charge of membrane pores..? The author need to be sure about their claims.
  • Line 207, stated that due to addition of nitric salt, NO3 are present in the feed solution. I think, the author need to understand the electro-neutrality conditions of given electrolytes. For an example, the concentration of Cu were prepared from Cu(NO3)2 so, the NO3 obviously present in the feed solution to maintain electro-neutrality condition of given electrolyte.
  • Line 212, ‘..reported to be at 6.0 ± 0.9 pH or to the point of zero charge..’ Why the author need to show this reference [25] on this manuscript. The way of study and membrane material are totally different on this reference number 25.  The claims and references need to be related with present study, so the author need to carefully to handle this section.
  • Line 216, the isoelectric point of given membrane can be determined by two methods like zeta potential measurments with different pH level or the rejection of given ions at different feed pH.
  • In general, when the isoelectric point (IEP) is located in acidic region, the membrane known to be as negative charge, while the IEP in base region the membrane should be as positive charge. In addition, the isoelectric point of given membrane should not change with the function of electrolytes. So the authors need to determine the isoelectric point of given Al2O3/TiO2 And it would helpful to evaluate the rejection coefficient of Zn2+, Cu2+, Fe3+ and Pb2+

What is the isoelectric point of Al2O3/TiO2 Ceramic membrane. ?

As per figure2, the isoelectric point of this membrane is located at pH 4.9, that mean the membrane is negatively charged.

  • Line 226, the author stated that, less than IEP of membrane the Cu2+ ions could be replled as per Donnan exclusion principle. Its correct.. !, and author also stated that, while above IEP the membrane is negatively charged, and it may adsorb the Cu2+ ions on the pore wall. Is it correct.. ? I think it’s Do you think, the higher rejection is used to occur on the principle of adsorption on membrane surface.. ? If the membrane adsorb cations, whether, the material balance is done at the end of study ?
  • And as per figure 3 to 6, the author used divalent cations (except Fe) on this study, however, the higher rejection used to occur at higher pH.. ? The author need to explain it.
  • As per Donnan principle, the higher rejection used to occur the repelling of ions on membrane surface. Is the author check the rejection % of NO3 at higher pH.. ? Because to maintain an electroeutrality condition, if NO3 is 90% rejection means Cu2+ would provide 90%. Again, is this membrane repel monovalent ions.. ?

How the volumetric charge density is higher on above IEP of membrane. The author mentioned (line279) divalent cations reduce the volumetric charge density as per adsorption prinicple.  Need more clarifications on this part, because the author didn’t account the dielectric exclusion as well as electrostatic intractions as per Donnan principle.

  • Line 280 -282, not required on this section and it may moved to introduction part.

Moreover, the author need to show the rejection mechanism of divalent cations on membrane surface. Because, the author stated that membrane have negatively charged and the volumetric charge density reduce due to divalent cations, however, the figures showed that, the higher rejection used to obtained at higher pH of feed solution.  First of all, the author need to calculate the isoelectric point of the desired membrane with respect to zeta potential value and it show whether the membrane is positively  charged or negatively.

Overall, the author did not or clearly mentioned about the mechanism of ions transport or rejection through the membrane with the function of volumetric charge density.  This manuscript need to revise it.

Reviewer 3 Report

The authors described the prediction of nanofiltration performance through DSPM model and verified the results with experiments. However, the current work lacks detailed discussion of this method, raising concerns about the controllability and reliability. Therefore, major revision should be requested before consideration of publication in Membranes.

  1. Equation 2 to 17 should be described one by one with the meaning of all variables.

  1. The DSPM should be described in detail for it’s the novelty of this research. And what’s the difference of the detailed and standard model?

  1. The calculated value of the charge density of the membranes should be compared with zeta potential of membranes.

  1. The authors selected four different ions with different molar concentration, what is the relevance of them. Do the results in this study have some reference value to other concentrations or ions?

Reviewer 4 Report

The manuscript ‘Impact of pH on total volume membrane charge density in nanofiltration of aqueous solutions containing heavy metals’ is a very interesting paper and has important practical aspects. I have a few comments that the authors should consider in preparing the revised version.

The Introduction section should be implemented with some references, e.g.:

  1. Kowalik-Klimczak, M. Zalewski, P. Gierycz, Removal of Cr(III) ions from salt solution by nanofiltration: experimental and modelling analysis, Polish Journal of Chemical Technology 18/3 (2016) 10-16.
  2. Kowalik-Klimczak, M. Zalewski, P. Gierycz, Prediction of the chromium(III) separation from acidic salt solution on nanofiltration membranes using Donnan and Steric Partitioning Pore (DSP) model, Architecture - Civil Engineering - Environment ACEE 9/3 (2016) 135-140.

In the Materials and Methods section, the experimental installation should be described, the membrane surface area should be provided and the method of conducting the process will be explained. In this section, the authors should also provide the time needed to obtain steady state. I am not quite clear why the ceramic membranes were used in this study. Please provide an explanation. Line 137, what is volume flux? Is it a permeate flux? Please check the correctness of the phrase used.

In the Results and Discussion section, it is necessary to extend the description of standard errors (Table 6).

Please update the literature. It is necessary to include the most recent literature reports in the manuscript.

English should be carefully revised through the whole manuscript. There are many mistakes in the manuscript related to the incorrect translation from Polish into English. Please correct the language according to the membrane terminology.

Round 2

Reviewer 2 Report

Manuscript ID: membranes-900186

Comments are given below

  1. The title is unclear, It mentioned as single solutions, however, it also stated that nitrate salts,

Hence, I request that, “title can be revised for reader’s better understanding with appropriate modification”

  1. In abstract:

The first sentence of abstract need to be revised, because the sentence is not complete properly.  it mentioned as "...investigated".. but it didn't mention as investigation on what basis..? and ..."using mathematical model..." it supposed to be mention for what purpose mathematical model used?.

In my suggestion, the sentence may re-frame like "The effect of pH on removal of nitric salts of Zn, Cu, Fe or Pb via nf membrane has been investigated by experimentally and DSPM model were used to determine the volumetric charge density of given electrolytes"

  1. All over the experimental studies, it clearly indicated that the isoelectric point (ISE) of given membrane was 4.9, because the retention coefficient was very less or near to zero. Moreover, the ISE is an inherent properties of membrane and it may not change with respect to different electrolytes. However, in figure 6, the ISE shown at 6.0. Is there any reaction between Al2O3/TiO2 membrane and Zn salts..? Whether solution pH played any role on isoelectric point of membrane? Could you please give some justification on this..?

  1. As per model, the volumetric charge density played a major role on ions rejection. It’s agreeable as per author’s justification. My question is how the volumetric charge density values (Xd) became positive on negative charge (base condition pH 7 - 9) of membrane..? Could you please justify it..?

Reviewer 3 Report

I think the paper can be accepted in present form.
